# The Hybrid Capture 2 Results in Correlation with the Pap Test, Sexual Behavior, and Characteristics of Romanian Women

**DOI:** 10.3390/ijerph20053839

**Published:** 2023-02-21

**Authors:** Toader Septimiu Voidăzan, Cosmina Cristina Uzun, Zsolt Kovacs, Florin Francisc Rosznayai, Sabin Gligore Turdean, Mihaela-Alexandra Budianu

**Affiliations:** 1Department of Epidemiology, George Emil Palade University of Medicine, Pharmacy, Sciences and Technology of Targu Mures, 540142 Targu Mures, Romania; 2Department of Biochemistry, Environmental Chemistry, George Emil Palade University of Medicine, Pharmacy, Sciences and Technology of Targu Mures, 540142 Targu Mures, Romania; 3Department of Obstetrics Gynecology, George Emil Palade University of Medicine, Pharmacy, Sciences and Technology of Targu Mures, 540142 Targu Mures, Romania; 4Department of Pathology, George Emil Palade University of Medicine, Pharmacy, Sciences and Technology of Targu Mures, 540142 Targu Mures, Romania

**Keywords:** cervical cancer, screening, human papillomavirus, hybrid capture 2 test

## Abstract

Background: Human papillomavirus (HPV) infection is the major cause of cervical cancer (CC); hence, it is critical to understand the processes by which HPV infection causes squamous intraepithelial lesions, as well as the proper diagnostic tools. The objective of this study was to establish the correlations between Pap test results and Hybrid Capture 2 (HC2) tests findings. Materials and methods: This study included 169 women between the ages of 30 and 64, who presented for consultation in gynecological clinics in both the public and the private sectors. These women reported symptoms, such as abnormal vaginal discharge and genital irritation, as well as early onset of sexual activity, multiple sexual partners, history of other sexually transmitted infections or high-risk sexual partners, immunosuppression, or tobacco smoking. Pap tests and HPV testing, using the HC2 method, were performed for the women enrolled in the study, including data gathered after patients completed questionnaires concerning their sexual behavior. Results: The HC2 method revealed that 66 patients (39.1%) tested positive for high-risk HPV types. Of the patients with positive results, 14 (21.2%) presented Atypical Squamous Cells of Undetermined Significance (ASC-US) compared to 10 (9.7%) patients in the negative group (*p* = 0.042). Atypical Squamous Cells for which a high-grade lesion cannot be excluded (ASC-H) were identified primarily in women with positive HC2 (6.1%). HR-HPV positivity was substantially more associated with low-grade ASC-US or low-grade squamous intraepithelial lesion (LSIL) and high-grade ASC-H cytology (OR = 2.53; 95% CI: 1.10–5.80, respectively, OR = 14.9; 95%CI: 1.006–34.59). Unmarried women (31.8%; *p* = 0.004) and women with multiple partners (over four partners, 10.6%; *p* = 0.03) were more likely to have an HPV infection when compared to married women and those with fewer sexual partners. Conclusions: Understanding the epidemiology of HPV genital infections is essential for developing preventive measures against this infection and CC. Identifying the most prevalent HPV types, and determining the incidence of HPV oncogenic infections, in conjunction with Pap test results and sexual behavior information, can constitute part of an algorithm for the efficient management of cervical intraepithelial lesions.

## 1. Introduction

Human papillomavirus (HPV) infection, the known cause of cervical cancer (CC), represents a subject of major interest in the medical literature. CC is the second-most-prevalent gynecological cancer worldwide. Almost all cervical precancerous lesions are caused by a chronic HPV high-risk oncogenic infection [1,2,3].

So far, more than 200 HPV types have been identified [4], 15 of those being high-risk (HR) (16, 18, 31, 33, 35, 39, 45, 51, 52, 56, 58, 59, 68, 73, 82), 3 classified as probably high-risk (26, 53, 66), and 12 low-risk (6, 11, 40, 42, 43, 44, 54, 61, 70, 72, 81, CP6108). HPV 16 and HPV 18 are the most frequent oncogenic types: HPV16 is implicated in 50–60% of CC and HPV 18 in 10–15%. The other types of HR-HPV are associated with 25–40% of the cases of CC [5].

It is estimated that nearly 80% of women will have an HPV infection throughout life, with an HPV oncogenic type causing 50–75% of those infections. Yet, more than 90% of cases are self-limited and will be gone in one or two years [6], with just a small proportion progressing to invasive CC [7,8].

Efficient identification of women with high-grade intraepithelial neoplasia (CIN3) during regular CC screening is a critical public health concern. Data available in the literature suggest that cervical cytology has a sensitivity that varies between 47% and 62%, with a specificity of 60% to 95% in detecting moderate and severe cervical dysplasia. In comparison to cytology, HPV genotyping by polymerase chain reaction (PCR) has a higher sensitivity but a significantly lower specificity, particularly in younger women [9].

The association between CC and HPV infection led to the development of different methods of cervical screening, especially HPV genotyping (DNA testing), used for identifying high-risk strains [10]. The disadvantage is that only one HPV-DNA detection cannot distinguish clinical persistent infection from transitory infection, making the search for specific markers for persistent HPV infection mandatory.

Hybrid Capture 2 (HC2) HPV-DNA reaction determines the HPV strains, this being a triage test used to separate the cases of low-grade cervical dysplasia into two groups: high risk and low risk for developing CC [11]. The HC2 HPV-DNA test is designed to augment existing methods for the detection of cervical disease and should be used in conjunction with clinical information derived from other diagnostic and screening tests, physical examinations, and full medical history following appropriate patient management procedures [12,13]. The HC2 HPV DNA test is used as a screening test in some national CC screening programs [14,15,16,17].

Although women in Romania have access to CC screening and HPV vaccination, this country has the highest burden of deaths caused by CC in Europe. When compared to the average mortality rate in Europe due to CC of 3.4/%000, the rate of 10.83/%000 in Romania is alarming and highlights the need for intervention [5,18].

The objective of this study is to identify a correlation between the results obtained after a Pap test and those after HC2, a method used to determine high-risk HPV types, less known and used in Romania, associating, at the same time, this information with data extracted from questionnaires, as well as data regarding sexual behavior and history.

## 2. Materials and Methods

### 2.1. Study Population

The selection of patients was carried out over a period of 18 months (2021–2022), and according to the inclusion and exclusion criteria, the selection was performed in three gynecological clinics (two from the public sector and one from the private sector), where the patients presented for gynecological consultation.

### 2.2. Inclusion and Exclusion Criteria

Inclusion criteria: women between the ages of 30 and 64, currently or previously sexually active, asymptomatic or with symptoms, such as abnormal vaginal discharge and genital inflammation, or with risk factors, such as the early onset of sexual activity, high-risk sexual partners, multiple sexual partners, history of other sexually transmitted infections, immunosuppression, and cigarette smoking.

Exclusion criteria: women under 30 years and over 64 years old, with a history of hysterectomy, with invasive surgical procedures in the genital sphere 12 months before, pregnant, or postpartum women.

A group of 200 eligible women was formed based on the budget allocated for this study for individual Pap tests and HPV genotype detection using the HC2 reaction. After excluding those who did not fill out the medical questionnaire, in the end, the study was carried out on 169 patients.

Patients included in this study gave their informed consent and were given information regarding the study protocol, which comprised cytological sample and biological collection, Pap test, and HPV genotype detection, using the HC2 reaction, results, etc.

The women filled out a short medical questionnaire and received the results and details about their health status by telephone. The results and all data were confidential. After selecting the patients, the gynecologist took cervical samples with a cervical brush using a liquid medium, using one sample for the Pap test and one for the HPV test, using the HC2 method. Following collection, cervical samples were stored at a temperature ranging from 2 °C to 8 °C, before being transported and processed by a pathologist. Following the examination of the cytological specimen using the Babeș–Papanicolaou testing method, the pathologist provided two sets of information, one related to premalignant/dysplastic and malignant lesions, being the cytopathological examination, and the second one referring to the microorganisms that can be identified by the method. Thus, the gynecologist can decide on HPV testing and colposcopy with biopsy addressed to the first category of lesions, while also deciding on antibiotic/anti-inflammatory treatment for infections.

### 2.3. Cytopathological Technique

Using the Bethesda system, the classification for cervicovaginal lesions was as follows: negative for intraepithelial lesions or malignancy (NILM), atypical squamous cells of undetermined significance (ASC-US), atypical squamous cells for which a high-grade lesion cannot be excluded (ASC-H), low-grade squamous intraepithelial lesion (LSIL), and high-grade squamous intraepithelial lesion (H-SIL). ASC-US describes an abnormal histological aspect of squamous cells, which usually indicates inflammation, reactive and reparatory alterations, or a precancerous lesion caused by persistent HPV infection. ASC-H describes the same abnormality, but the pathologist cannot exclude a high-grade lesion [19,20].

Microbiological evaluation of vaginal discharge revealed possible infections caused by different microorganisms, such as *Candida*, *Trichomonas*, *Gardnerella*, *Chlamydia trachomatis*, *Mycoplasma*, etc. The identification of microorganisms was carried out on the cytological specimens.

### 2.4. Detection of HPV Types

For the detection of high-risk HPV, we used the HC2 assay (Qiagen). The HC2 analysis was performed according to the manufacturer’s recommendations, with results being positive or negative using 1.0 relative light units/cut-off (RLU/CO) as the cut-off value [13,14]. We considered the results to be positive when they exceeded 1.0 RLU/CO.

The HC2 technique of DNA-based testing for high-risk HPV is a nucleic acid hybridization test with signal amplification, employing chemiluminescence of microplates for quality detection of 13 high-risk types of HPV: 16, 18, 31, 33, 35, 39, 45, 51, 52, 56, 58, 59, and 68. The samples were collected and transported using the test kit’s medium. We were able to retain the samples at room temperature for two weeks, before storing them at 2 °C to 8 °C, for one week. If testing was performed after more than three weeks, samples were stored at −20 °C. The DNA was neither altered nor contaminated as a result of this technique during transit and storage. HPV testing services were performed at Oncological Institute “PROF. Dr. I. CHIRICUŢĂ” Cluj-Napoca, based on a collaboration protocol.

Using a medical questionnaire, selected patients were evaluated based on age, economic–social status, education level, marital status, tobacco smoking, alcohol consumption, sexual behavior, and characteristics (menarche, age when first sexual intercourse took place, number of sexual partners, history of sexually transmitted infections), personal physiological antecedents (numbers of pregnancies, births, and abortions), and history of illnesses.

This questionnaire-based study was approved by The Ethics Committee of George Emil Palade University of Medicine, Pharmacy, Science, and Technology of Targu Mures (no. 530/21/11/2019).

### 2.5. Statistical Analysis

Data were collected in Excel spreadsheets and statistically analyzed using the Statistical Package for Social Science (SPSS, version 23.0.0, Chicago, IL, USA).

Absolute and relative frequencies (%) were used to represent nominal variables, and the association between these variables was analyzed using Pearson’s Chi-square test or Fisher’s exact test. The size effect was reported as an odds ratio (OR) with a 95% confidence interval for the statistically significant associations, and with *p* < 0.05. All *p* values reported are two-tailed.

## 3. Results

### 3.1. Correlation between the HC2 Results and Pap Test Results

The study included 169 patients, and the HC2 method revealed that 66 patients (39.1%) tested positive for high-risk HPV types, whereas 103 patients (60.9%) tested negative. Of the patients with positive results, 14 presented ASC-US (21.2%), compared to 10 patients in the negative group (9.7%) (*p* = 0.042). ASC-H was identified primarily in women with positive HC2 (6.1%), while NILM was found mostly in women with negative HC2 (33.3%). The cytopathological type H-SIL was not identified in those patients. The colposcopy/biopsy recommendation was noted on the cytology exam bulletins in 18.2% of the patients with positive HC2 tests (*p* = 0.0001). Among the microorganisms, *Candida albicans* and *Gardnerella vaginalis* were identified (Table 1). HR-HPV positivity was substantially more associated with low-grade ASC-US or LSIL and high-grade ASC-H cytology (OR = 2.53; 95% CI: 1.10–5.80, respectively, OR = 14.9; 95%CI: 1.006–34.59).

### 3.2. Correlation between the HC2 Results and Sexual Behavior and Characteristics

Table 2 shows the impact of sexual behavior on HC2 outcomes. The risk of a positive result is higher amongst unmarried women than married ones, with 31.8% vs. 13.6% (*p* = 0.004). More than 60% of patients, regardless of their findings, believe that the first sexual encounter or first time living with a partner should occur after the age of 18. More than 90% of patients had a consistent sexual partner, and 10.6% of the women with positive results had at least four sexual partners up to the day of inclusion in the study (*p* = 0.03). The distribution of the two groups was similar in response to the following questions: “*Have you had sexual contact outside your couple?*”, “*Duration of sexual activity*”; “*Frequency of sexual acts per week or at what age does sex should usually start?*” A difference was observed between the groups in terms of extramarital sexual intercourse, with 13.6% of the positive patients requesting a consultation with a specialist doctor after 4 to 6 months (*p* = 0.04). Bleeding accounted for 50% of the positive patients, indicating that they should visit a doctor for a specialist consultation (*p* = 0.038). For both subgroups of patients, doctors/health workers and the internet are the two primary sources of information about sexually transmitted illnesses (Table 2).

### 3.3. Correlation between the HC2 Results and the Information Regarding HPV Infection

Although the most of the patients enrolled in this study have heard about HPV infection and are familiar with HPV-related disorders, there was a significant percentage of incorrect transmission methods described by the patients (unsterile needles or blood transfusion/organ transplant). Further, 81.8% of the women with positive results considered that HPV infection is a high risk for their health status (*p* = 0.02). Women in both groups responded similarly to the method of prevention/reduction in HPV infection and awareness about HPV, as well as the number they received from a Pap test. When evaluating the causes for the patients not undergoing a Pap test, the lack of time and fear of a positive result were the primary reasons, but a significant number of women considered that this test is unnecessary. The two main sources of information regarding HPV infection are medical doctors and the internet, and patients consider that school and medical doctors should offer more information regarding HPV infection (Table 3).

## 4. Discussion

This study aimed to evaluate the diagnosis of intraepithelial cervical lesions using the standard investigation method, the Pap test, and the modern HC2 method, as well as to correlate these techniques with a variety of risk factors, which were assessed using a survey questionnaire to women enrolled in the study.

HPV infection is an important precondition, but not enough for developing CC. Numerous cofactors were identified: first sexual intercourse at a young age, infections caused by Herpes simplex virus type 2 or *Chlamydia trachomatis*, frequent change of sexual partners, long-term use of oral contraceptives, tobacco smoking, a diet low in fruits and vegetables, obesity, multiple pregnancies, giving birth before the age of 17, and a family history of CC [21].

The Pap test is used for cervical screening and can help reduce CC mortality when associated with prevention methods and treatments. However, this test has a low sensitivity, high false-negative rates (30%), and significant false-positive rates (between 15% and 50%) [22]. Using HPV-DNA testing for screening improved the outcomes of the cytological triage [23].

Establishing the diagnosis algorithm is not complete without the detection of HPV types. Using manufacturer’s recommended cut-off (RLU/CO = 1.0), sensitivity was 93.8% (95% CI: 90.1–97.5) and specificity was 63.2% (IC 95%: 58.6–67.6%) [14].

In our study, the percentage testing positive for high-risk HPV types at HC2 was 39.1 (95% CI: 30.2–49.7%), with most of the women presenting symptoms, such as abnormal vaginal discharge, genital inflammation, or one of the risk factors known to be associated with a higher risk for developing CC. In the group of women with positive HPV test via HC2, when evaluating low-grade lesions (ASC-US + LSIL), we identified a percentage of 22.7%, compared to 12.6% in the group with a negative test. When evaluating women with a positive HPV test by HC2, 6.1% had a high-grade lesion (ASC-H), compared to 0.0% amongst women with a negative HPV test. After calculating the causal relationship between cytology and positive HR-HPV using HC2, we obtained an OR = 2.53; 95%CI: 1.10–5.80 for positive HPV and low-grade cytology (ASC-US or LSIL), and an OR = 14.9; 95%CI: 1.006–34.59 for positive HPV and high-grade cytology (ASC-H). We, thus, mention that there is a correlation between HC2 positivity and high-grade cytological lesions.

According to Oliviera et al. [24], HR-HPV was confirmed positive in 64.39% of ASC-US and 65.38% of ASC-H. Optimal ASC-US triage should identify patients who require treatment while minimizing needless procedures. However, there is a disagreement over the most effective strategy for preventing ASC-US from evolving to higher-grade lesions [25].

Romania has a high incidence and mortality rate caused by CC [5,26], with very little data regarding the prevalence of HR-HPV infection. According to a study by Mihaela et al. [26], the incidence of HPV infection among women between the ages of 18 and 59 is 40% for all types of HPV, and 20% for HR-HPV. Further, Ilisiu [27] reports a prevalence of 16.9% in the general population (IC 95%: 14.7–19.5%), with Poland at 16.6%, the Czech Republic at 18.2%, Slovenia at 12.9%, and Greece at 12.7%. The global prevalence of HPV infection among women without cervical abnormality is 11.0% to 12.0%, with Sub-Saharan Africa at 25.0%, Eastern Europe at 21.0%, and Latin America at 16.0%, having the highest rates [3].

In a study conducted by Agorastos et al. [28], a prevalence of 12.7% was reported for HR-HPV, 2.7% for HPV-16, and 1.4% for HPV-18. Furthermore, in another study, low-grade cytology (ASC-US or LSIL) and high-grade cytology (ASC-H or HSIL) were strongly associated with a positive HR-HPV (OR = 3.17; 95% CI: 1.65–6.08, and OR = 5.18; 95% CI: 2.74–9.79), women with previous Pap tests being more likely to have a positive HR-HPV (OR = 2.54; IC 95%: 1.89–3.41) than those with no previous testing [27].

In our study, women with abnormal cytology and positive HPV test were advised to follow up and to repeat the test in 6 months and/or colposcopy and biopsy (18.2%; *p* = 0.0001). Additionally, the correlation between HC2 results and sexual behavior/characteristics was evaluated. Among patients with a positive result, it was shown that unmarried women (31.8%; *p* = 0.004) and those with multiple sexual partners (more than 4 = 10.6%; *p* = 0.03) have a higher risk of HPV infection when compared to those married or with fewer sexual partners. The study highlighted that 13.6% of positive women came for a gynecological consultation 4–6 months after having extramarital sexual intercourse (*p* = 0.04), knowing better than patients with negative results that bleeding is a symptom that would determine them to ask for a consultation (*p* = 0.038).

According to Rada C [29], in a study on sexual behavior in Romania, with 1200 sexually active participants, 34.8% had their first sexual intercourse between the ages of 18 and 19, and 30.1% of them between the ages of 16 and 17. More than 60% were not in a stable relationship with their sexual partner, and only 23.7% used protection for their first sexual encounter, the average number of sexual partners being 4.48. From those included in the study, 44.2% received information about sex after the age of 15, from friends or acquaintances (35.2%), and mass media or the internet (26.4%), these reported as the main sources of information regarding sexual behavior.

In a study by Ribiero et al. [30], which included 198 women, 32.3% had their first sexual intercourse before the age of 16, 20.2% had only one sexual partner, and 79.8% had two or more, with the majority of women (84.3%) having at least one pregnancy, with oral contraceptives reported to be used by 24.7% of women. Further, among women with abnormal cervical cytology, 87% of those had an HPV infection, with 42.4% presenting type 16 and type 18, and 43.9% with other HPV strains.

Other epidemiological studies show the same association between behavioral risks, such as age at first sexual intercourse, number of sexual partners and the partner’s sexual behavior, and an HR-HPV infection [31,32,33].

The correlation between HC2 results and knowledge regarding HPV infection showed that most of the patients (over 85%) had heard about HPV but did not know what diseases are associated with this infection; a high proportion of these patients highlighted incorrect ways of contracting the infection, such as unsterilized needles (28%) or blood transfusion/organ transplant (27%). Among women with positive results, 81.8% consider HPV infection to have a high risk for health status, compared to 64.1% in the group with a negative result. When asked about methods of prevention/reductions in HPV infection or knowledge about HPV and how many times they undertake a Pap test, it was shown that lack of time and fear of a positive result are the main reasons, but 56.1% of the positive group and 61.2% from the negative group consider a Pap test as not being necessary.

According to GFK research [34,35], conducted in 2016, 1 in 10 women in Romania did not get a medical routine consultation in the previous 10 years, with 35% saying they go less than once every three years, 7 out of 10 women had no test to identify precancerous lesions or HPV infection, with only 23% of women having a Pap test in the last 3 years, and just 5% having both a Pap test and a DNA-HPV detection. This issue has been reviewed in a cross-sectional study that used a structured questionnaire, showing that the main reason for women not participating in screening campaigns is because they do not know of the programs or high costs [31].

According to our findings, the two main sources of sexually transmitted disease information are medical workers (59.1% vs. 57.3%: positive group vs. negative group), and the internet (43.9% vs. 39.8%: positive group vs. negative group). The same applies to information regarding HPV infection, doctors (71.2% vs. 74.8% positive group vs. negative group), and the internet (51.5% vs. 43.7% positive vs. negative). In the opinion of patients, doctors (95.5% vs. 75.7% positive group vs. negative group) and schools (42.4% vs. 44.7% positive group vs. negative group) should make information about HPV infection available. Further, in a previous study, by Voidăzan et al. [5], it was shown that the majority of the respondents obtained information about HPV infection and vaccination from the internet, with only a small proportion searching information at their general practitioner or specialist, considering that doctors should make information on HPV available.

Women’s adherence to screening programs and medical dissemination of relevant information are important factors in reducing the risk of HPV-associated cancer, and campaigns to inform the population about screening and vaccination programs represent effective ways to reduce incidence, mortality, and morbidity. In Romania, vaccination against HPV is not included in the National Vaccination Plan; instead, it is free and is carried out in family doctors’ offices, for females aged 11 to 18, based on parental request.

In Romania, the first regional screening program for CC was implemented in 2002, in the northeastern region of the country, with a coverage area of 21% [36]. In June 2012, the Romanian Health Ministry [37] started a national screening program for CC for a period of 5 years. Using the Pap test, screening for CC was performed for women between the ages of 25 and 64, without symptoms or history of CC and regardless of insurance status. Women presenting the absence of a cervix, congenital or post-surgery, and with a history of genital cancer were not eligible. Those above the age of 64 with negative results from the previous three Pap tests were exempt from testing.

Nowadays, programs for the early detection of CC using the Pap test are in place in Romania. The goal of this screening program is to reduce the burden of CC by detecting it in its early stages, guiding patients with precancerous lesions toward specialized diagnostic and treatment centers, educating the population regarding screening as a method for early detection of CC in asymptomatic women.

The follow-up process for women with squamous intraepithelial lesions is rather laborious and with extra costs that are covered by the national screening program. For ex-ample, HPV-DNA testing is included in the programs for prevention, early detection, diagnosis, and prompt management of CC.

This type of program has the benefit of reducing poverty and any kind of social exclusion, allowing access to a high-quality healthcare system and sustainable social services. At least 50% of women are part of vulnerable groups. A screening program consists of interventions and complex measures that require a vast mobilization of financial, human, and material resources in order to attain high performance rates in accordance with European standards.

In order to enhance the degree of acceptance for our screening methods among women in Romania, efforts must be made to: (1) encourage eligible women to take the test as part of our national screening program, hoping that after, they will start taking their routine medical appointments and learning basic rules of hygiene and healthy lifestyle; (2) let people know that screening for CC is free; (3) create an active role of healthcare professionals in promoting information about sexual health, discussing with younger women who have a higher risk of acquiring HR-HPV infection, facilitating access to screening in isolated communities from rural areas [28].

The limitation of this study is the absence of probability sampling when enrolling the patients, given that patients were enrolled from public and private clinics where they attended routine check-ups. However, the research population’s diverse socio-demographic characteristics, the pre-established inclusion criteria, and the high rates of response to our surveys alleviate this susceptibility. Another limitation is from the method of completion; given that the questionnaire was self-managed, perhaps some of the items were not comprehended, which draws into doubt the reliability of their answers. To alleviate this constraint, all incomplete questionnaires were excluded after conducting an in-depth review of the study’s internal validity.

## 5. Conclusions

Understanding the epidemiology of genital HPV infection is essential for developing preventive measures against this infection and HPV-related neoplasia. By identifying the most frequent HPV types and calculating the incidence of HR-HPV infections, based on Pap test results and the patient’s sexual activity, an algorithm can be established for better management of women with cervical intraepithelial lesions. Doctors can assist with reliable information regarding the diagnosis method, facilitating access to effective and prompt cervical dysplasia therapy.

Any screening program’s primary goal is to minimize morbidity and mortality. Long term, in order to reduce the mortality caused by CC, changes in the behavior of the population must be undertaken, promoting health actions and reducing the increased risk of the disease, prolonging an active life, and reducing the incidence of CC and the associated complications. By enhancing the interest in preventive services, we can teach the population that they need to prevent, not just treat, diseases.

## Figures and Tables

**Table 1 ijerph-20-03839-t001:** The HC2 results in correlation with the Pap test results.

Variables		HC2 Result		*p* Value
		Positive 66 (39.1%)	Negative 103 (60.9%)	
Squamous cells	**ASC-US**	14 (21.2)	10 (9.7)	**0.042**
	ASC-H	4 (6.1)	0 (0.0)	**0.04**
	L-SIL	1 (1.5)	3 (2.9)	0.94
	NILM	9 (13.6)	34 (33.3)	**0.007**
	No epithelial cell abnormalities	38 (57.6)	56 (54.4)	0.80
Non-cancerous alterations	Mature squamous metaplasia	20 (30.3)	43 (41.7)	0.18
	Inflammation	21 (31.8)	37 (35.9)	0.70
	Atrophy	0 (0.0)	1 (0.9)	0.76
	No non-neoplastic changes	29 (43.9)	29 (28.1)	0.05
Recommendations	Colposcopy/Biopsy	12 (18.2)	0 (0.0)	**0.0001**
	Repeat after 6 months	10 (15.6)	17(16.4)	0.52
	Repeat after treatment	22 (33.3)	33 (32.0)	0.95
	Routine check	22 (33.3)	53 (51.5)	0.03
Microorganisms	*Candida albicans*	10 (15.2)	9 (8.7)	0.29
	*Gardnerella vaginalis*	1 (1.5)	8 (7.8)	0.15
	No microorganisms	55 (83.3)	86 (83.5)	0.87

**Table 2 ijerph-20-03839-t002:** Results of the HC2 testing in correlation with sexual behavior and characteristics of patients.

Variables		HC2 Result		*p* Value
		Positive 66 (39.1%)	Negative 103 (60.9%)	
Marital status	**Married**	39 (59.1)	75 (72.8)	**0.0001**
	Living with partner	3 (4.5)	4 (3.9)	0.83
	Divorced	3 (4.5)	10 (9.7)	0.34
	Unmarried	21 (31.8)	14 (13.6)	**0.004**
First sexual intercourse	Before the age of 18	22 (33.3)	45 (43.7)	0.23
	After the age of 18	44 (66.7)	58 (56.3)	0.23
Safe first sexual intercourse	Yes	40 (60.6)	52 (50.5)	0.25
First time living with partner	Before the age of 18	7 (10.6)	20 (19.4)	0.19
	After the age of 18	45 (68.2)	77 (74.7)	0.45
	NA	14 (21.2)	6 (5.8)	**0.005**
Do you have a stable relationship?	Yes	61 (92.4)	95 (92.2)	0.95
How many sexual partners did you have until today?	1	16 (24.2)	35 (34.0)	0.23
	2	36 (54.7)	51 (49.5)	0.49
	3	7 (10.6)	15 (14.6)	0.60
	4	7 (10.6)	2 (1.9)	0.03
Did you have sexual intercourse outside your stable relationship?	Accidentally	6 (9.1)	8 (7.8)	0.98
	More than once	2 (3.0)	2 (1.9)	0.95
	From time to time	1 (1.5)	8 (7.8)	0.15
	Never	57 (86.4)	85 (82.5)	0.64
Duration of sexually active life	Less than 10 years	15 (22.8)	21 (20.3)	0.84
	11–20 years	38 (57.6)	53 (51.5)	0.53
	More than 21 years	13 (19.7)	29 (28.2)	0.28
Frequency of sexual intercourse/week	1	41 (71.9)	71 (63.4)	0.32
	2	14 (24.6)	34 (30.4)	0.52
	3	1 (1.8)	2 (1.8)	0.55
	4	1 (1.8)	5 (4.5)	0.61
At what age do you consider appropriate beginning the sexual life?	14–17 years	13 (22.8)	37 (33.0)	0.21
	18–21 years	43 (75.4)	72 (64.3)	0.17
	22–25 years	1 (1.8)	2 (1.8)	0.55
	Older than 25 years	0 (0)	1 (1.8)	0.73
At what age did you receive the first informative material about sex?		14.75 ± 1.91	14.49 ± 2.05	0.40
How much time have passed after sexual intercourse outside your stable relationship and a gynecological consultation?	Less than 2 months	5 (7.6)	5 (4.9)	0.69
	2–4 months	0 (0)	1 (1.8)	0.73
	4–6 months	9 (13.6)	4 (3.9)	**0.04**
	NA	52 (78.8)	93 (90.3)	0.06
What symptoms made you/can make you take a gynecological consultation?	Itches	24 (36.4)	28 (27.2)	0.27
	Abnormal vaginal discharge	38 (57.6)	49 (47.6)	0.21
	Genital lesion	22 (33.3)	22 (21.4)	0.10
	Bleeding	33 (50.0)	35 (34.0)	**0.038**
Which are the two sources of information regarding sexually transmitted diseases?	Medical doctors, nurses	39 (59.1)	59 (57.3)	0.81
	Parents, family	23 (34.8)	28 (27.2)	0.29
	Friends, acquaintances	18 (27.3)	29 (28.2)	0.90
	Internet	29 (43.9)	41 (39.8)	0.59
	Books, magazines	17 (25.8)	19 (18.4)	0.33
	School	0 (0)	6 (5.8)	**0.046**
	Newspaper, radio, TV	3 (4.5)	9 (8.7)	0.30

**Table 3 ijerph-20-03839-t003:** HC2 results in correlation with knowledge about HPV.

Variables		HC2 Result		*p* Value
		Positive 66 (39.1%)	Negative 103 (60.9%)	
Did you know about HPV infection?	Yes	60 (90.9)	88 (85.4)	0.34
How can you acquire an HPV infection?	Heterosexual intercourse	50 (75.8)	73 (70.9)	0.82
	Homosexual intercourse	37 (56.1)	48 (46.6)	0.46
	Intimate touch	18 (27.3)	18 (17.5)	0.22
	Unsterile needles	19 (28.8)	33 (32.0)	0.94
	Blood transfusion/organ transplantation	18 (27.3)	28 (27.2)	0.99
Do you consider that HPV is a risk for your health status?	High risk	54 (81.8)	66 (64.1)	**0.02**
	Medium risk	6 (10.3)	20 (19.4)	0.17
	Low risk	1 (1.8)	2 (1.8)	0.55
	I don’t know	5 (7.6)	15 (14.6)	0.25
Which are the diseases caused by HPV?	CC	54 (81.8)	82 (79.6)	0.23
	Anal cancer	15 (22.7)	16 (15.5)	0.15
	Oral cancer	7 (10.6)	25 (24.2)	**0.034**
	Genital condylomatosis	25 (37.9)	34 (33.0)	0.33
Choose the methods for prevention/reduction of HPV infection	One sexual partner	13 (19.7)	24 (23.3)	0.71
	Using the condom	16 (24.2)	21 (20.4)	0.68
	Vaccination	11 (16.7)	17 (16.5)	0.37
	Personal hygiene, soap, and water after sexual intercourse	5 (7.6)	5 (4.9)	0.69
	Pap test	9 (13.6)	20 (19.4)	0.57
Knowledge about HPV	Most of the HPV infection need medical treatment	21 (31.8)	37 (35.9)	0.20
	CC or dysplasia treatment eradicates the infection	16 (24.2)	16 (15.5)	0.30
	Genital condylomatosis is caused by the same HPV types that causes CC	18 (27.3)	29 (28.2)	0.68
	Most of the cases of CC are caused by HPV infection	36 (54.5)	55 (53.4)	0.42
	Women with HPV infection cannot receive the vaccine	18 (27.3)	22 (21.4)	0.32
	Did you receive a vaccine against HPV infection?	3 (4.5)	6 (5.8)	0.30
How many times did you undertake a Pap test?	Never	8 (12.1)	6 (5.8)	0.24
	2–3 times	18 (27.3)	27 (26.2)	0.95
	More than 3 times	40 (60.6)	70 (68.0)	0.41
What made you not to go for a Pap test?	Time	15 (22.7)	20 (19.4)	0.74
	Money	7 (10.6)	4 (3.9)	0.16
	Fear of a positive result	7 (10.6)	16 (15.5)	0.49
	I considered that is not necessary	37 (56.1)	63 (61.2)	0.61
Which are the two main sources for information regarding HPV infection?	School	7 (10.6)	8 (7.8)	0.52
	Books, magazines	8 (13.6)	17 (16.5)	0.61
	Parents	5 (7.6)	16 (15.5)	0.12
	Family	5 (7.6)	15 (14.6)	0.13
	Internet	34 (51.5)	45 (43.7)	0.32
	Doctors	47 (71.2)	77 (74.8)	0.61
In your opinion, who should offer more information regarding HPV infection?	School	28 (42.4)	46 (44.7)	0.77
	Books, magazines	9 (13.6)	23 (22.3)	0.16
	Newspaper, radio, TV	11 (16.7)	25 (24.3)	0.24
	Doctors	63 (95.5)	78 (75.7)	**0.001**
	Internet	0 (0.0)	2 (1.9)	0.25

## Data Availability

Not applicable.

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
