# Peer review of "The Hybrid Capture 2 Results in Correlation with the Pap Test, Sexual Behavior, and Characteristics of Romanian Women"

_ijerph, 2023, doi:10.3390/ijerph20053839_

Round 1

Reviewer 1 Report

The present work looks for the relationship between the Pap and the capture of hybrids, I have some points that need to be addressed.

* In the intro they don't mention anything about capturing hybrids, what is it? What does it identify? In other countries is it used as a diagnosis or not?

In the results section:

1. Correlation between HC II results and Pap smear results

*Although they describe how many samples were positive and negative, they do not mention which of these were associated with the Pap smear. I think it should be clear in the description of the results because it is the main result.

*They mention that they determined the confidence intervals (CI) of the odds ratios (ORs), but they do not mention in the statistics section what type of analysis they used and what the confidence interval is, for example: odds ratios (ORs) and 95% confidence intervals (CI) were calculated using univariate logistic regression (unadjusted model).

*In the discussion they do not mention whether or not there is a correlation between the pap and the capture of hybrids, nor do they mention other works on this data.

*In paragraph 204 and 205 they mention: In the group of women with positive HPV test, when evaluating low- 204 grade lesions (ASC-US or LSIL) we identified a percentage of 22.7, compared to 12.6% in 205 the group with negative test. Was this result positive in Pap and what happened with the capture of hybrids was it positive or negative? it is not clear.

*I understand that the study population was Romania, in the introduction they do not mention anything about how the prevalence of HPV or CC is in this place (despite the fact that there are already vaccines), what is the diagnostic method used for infection by HPV and CC?. They should state that HPV and CC is a health problem in Romania and therefore studies are necessary to make timely and accurate detection and prevent deaths.

*In the discussion they mention the results obtained from sexual behavior and mention other works, but they do not compare with their data, if there is a relationship or not with those works, or if there are differences because they believe that it is the difference, for example, the type of population , the number of samples, etc.

*In the discussion at the end they mention the health problem and what to do to help the population is a very good point. It may be important to mention what happens with vaccination in Romania.

Reviewer 2 Report

In the manuscript by Voidazan et al the authors describe a study examining sexual behavior characteristics of women in relation to HC2 HPV results. The study was performed in Romania that apparently still struggles with cervical cancer screening programs and the current study will raise awareness and hopefully contribute towards eliminating this cancer.

However, the methods section as well as some of the results are suboptimal and leave a lot unsaid. This uncertainty can affect the conclusions. For example 40% HPV positivity in general screening population implies that the study population is not adequately described. Furthermore Table 1 contains many “NA” values and if indeed those are missing values /”not analyzed” this would mean that the data is not robust. If more than 50% of patients have no cytological diagnosis which is the basis of cervical cancer screening how reliable is the rest of the data collected?

The manuscript could benefit from some language revision even though the main message can be understood.

Abstract lacks total number of enrolled participants.

P2 L74 Section about the study population doesn’t provide relevant information like when the collection took place, where the collection took place. How many were invited/eligible/included. The inclusion criteria are quite wide and without further information it is difficult to say whether the finally collected 169 is strangely low or not. On the other hand almost 40% HPV positivity in something described as “routine gynecological consultation“ is a bit unusual and might warrant more detailed description

P2 L88 „medium of origin“ unclear

P2 L95 the first sentence of the paragraph seems obsolete since the aim was not to assess cost effectiveness ratio

P3 L97 not enough methodological information was provided for microbial detection. At least the microorganisms found in the results should be listed in methods (table 1)

P3 L100 some meaningful break should be made before cervical lesion grading. Currently the text reads as if Bethesda system was relevant for bacteria.

P3 L117 typo in temperature notation „one-week storage at 2-80C. In the event of testing after more than three weeks, samples were stored at -200C“

P3L123-125 name of the statistical package is repeated unnecessarily

P3 L144 table 1 should not be split across 2 pages

P3 L144 table 1 what is the NA in microorganisms and cytological diagnosis?

P7 L179 the first sentence is not very related to the manuscript. Herein neither the mechanisms of lesion causation nor methods are investigated.

P8 L195 typo HPv-DNA

Reviewer 3 Report

The authors presented the results of Hybrid Capture II in correlation with Pap smear and sexual behavior and characteristics. It could be worthy for clinical application, especially for women living in HPV testing not convenient area. I have suggestions as follows. 

1.Several abbreviations in abstract section had no detailed description, such as (ASC-US or LSIL) and (ASC-H or HSIL) on Line 32. 

2.In introduction section, it is not clear “Worldwide CC is the second most frequent cause of genital cancer” and no appropriate reference(s) on Line 47-48.  

3. In section of Materials and Methods, authors described “Using a medical form, selected cases were evaluated depending on age, medium of origin, economic-social status, education level, marital status, cigarette smoking, alcohol consumption, sexual behavior and characteristics ---.” Some factors were not included for calculating to show the basic characteristics related to the risk of cervical cancer risks. As this study involving questionnaire about knowledge HPV infection, education level of individual may be an important factor in the study. Authors can discuss this point.

Reviewer 4 Report

This is a study that investigates the relationship between an HPV test (Hybrid Capture II), cytopathologic findings (Pap smear) and other clinical characteristics such as sexual behavior and knowledge. The following can help improve this paper:

MAJOR

1. The title should be revised and improved. Suggestion: Correlation of DNA-based HPV testing and clinicipathologic characteristics of Romanian women.

2. The manufacturer of the test kit and other equipment utilized in Hydrid Capture II must be indicated.

3. Explain why there was no HSIL in this cohort and only ASC-H.

4. Show photos of the cytopathology findings for all categories.

5. Statistical analysis: Please indicate the exact statistical test used.

6. Please define the "positive" and "negative" result of the hybrid capture II

7. Please indicate the source/reference of the questions asked to analyze knowledge.

8. Please shorten your discussion and just address the aims, objectives and findings of your investigation.

MINOR

1. Extensive spelling and grammatical errors are present. Please review and edit.

2. Degrees Celsius was indicated as zeros. Please correct.

Round 2

Reviewer 1 Report

The paper improved 

Author Response

Thank you very much for appreciating the article. Respectfully! Authors

Reviewer 2 Report

The major comments were adequately addressed

the manuscript appears technically sound.

Hopefully the readers will understand that this is not a routine gynaecological screening population but likely inherently biased population towards "interesting cases" hence the high HPV prevalence

Author Response

(The authors gave the same response as above.)

Reviewer 4 Report

The changes in this version have greatly improved the presentation and readability of the paper. However, the following are suggested:

1. In the introduction, please include the following papers on HPV and cervical cancer demographics: PMID 26126623, PMID 23199955

2. Materials and Methods 

- Please specify IRB approval if secured.

- In the inclusion criteria, the presence of symptoms need not be indicated. If this is not a required element to include a patient, then just describe the symptoms under results.

- Under cytopathological technique, please clarify if histologic correlation was performed. If not, please exclude lines 133 to 136.

- Was microbiological evaluation using culture performed in all specimens? Please clarify and show results. I assumed that the identified microorganisms was done by cytopathology as well.

- Please define what RLU/CO means.

- Please separate lines 154-161 in a separate heading (clinical data).

- For statistical analysis, can the authors explain why an older version of SPSS was used? Its latest version is now version 29. Please explain if all p values were two-tailed or one-tailed. 

3. Results

In results, line 180 please do not use "germs"

- Lines 2-5-206, what does this question/statement have to do with this study: "more than 60% of patients, regardless of their findings, belive that first sexual encounter or first time living with partner occur after the age of 18"? This "belief" seems unnecessary. Similarly, the discussion on extramarital affair and question on "stable relationship" is not appropriate. I wonder how the questionnaire was approved by the IRB when these seemingly intrusive and non-scientific questions were included.

- Please clarify how the source of information regarding HPV infection is relevant for this study?

4. Dicussion

- Please clarify lines 373-375. Is vaccination against HPV free in Romania?

- Please improve the discussion based on the aims of the study.

5. The title needs grammatical correction.

6. The entire document requires extensive English editing.

Author Response

(The authors gave the same response as above.)
